# Elevated Expression of JMJD5 Protein Due to Decreased miR-3656 Levels Contributes to Cancer Stem Cell-Like Phenotypes under Overexpression of Cancer Upregulated Gene 2

**DOI:** 10.3390/biom12010122

**Published:** 2022-01-12

**Authors:** Natpaphan Yawut, Il-Rae Cho, Phatcharaporn Budluang, Sirichat Kaowinn, Chutima Kaewpiboon, Byeoleun Jeon, Sang-Woo Kim, Ho Young Kang, Min-Kyung Kang, Sang Seok Koh, Young-Hwa Chung

**Affiliations:** 1BK21 Plus, Department of Cogno-Mechatronics Engineering, Optomechatronics Research Center, Pusan National University, Busan 46241, Korea; natpaphan21@gmail.com (N.Y.); irchohj@hanmial.net (I.-R.C.); phatcharaporn.bud@gmail.com (P.B.); 2Department of General Science and Liberal Arts, Ladkrabang Prince of Chumphon Campus, King Mongkut’s Institute of Technology, Chumphon 86160, Thailand; sirichat29@gmail.com; 3Department of Biology, Faculty of Science, Thaksin University, Patthalung 93210, Thailand; chutimak@gmail.com; 4Department of Biological Science, Pusan National University, Busan 46241, Korea; starsilver20@naver.com (B.J.); kimsw@pusan.ac.kr (S.-W.K.); 5Department of Microbiology, Pusan National University, Busan 46241, Korea; hoykang@pusan.ac.kr; 6Department of Biomedical Sciences, Dong-A University, Busan 49315, Korea; alsrud3692@naver.com (M.-K.K.); sskoh@dau.ac.kr (S.S.K.)

**Keywords:** cancer upregulated gene, Jumonji C domain-containing protein 5, cancer stem cell, miR-3656

## Abstract

Overexpression of cancer upregulated gene (CUG) 2 induces cancer stem cell-like phenotypes, such as enhanced epithelial-mesenchymal transition, sphere formation, and doxorubicin resistance. However, the precise mechanism of CUG2-induced oncogenesis remains unknown. We evaluated the effects of overexpression of CUG2 on microRNA levels using a microRNA microarray. Levels of miR-3656 were decreased when CUG2 was overexpressed; on the basis of this result, we further examined the target proteins of this microRNA. We focused on Jumonji C domain-containing protein 5 (JMJD5), as it has not been previously reported to be targeted by miR-3656. When CUG2 was overexpressed, JMJD5 expression was upregulated compared to that in control cells. A 3′ untranslated region (UTR) assay revealed that an miR-3656 mimic targeted the JMJD5 3′UTR, but the miR-3656 mimic failed to target a mutant JMJD5 3′UTR, indicating that miR-3656 targets the JMJD5 transcript. Administration of the miR-3656 mimic decreased the protein levels of JMD5 according to Western blotting. Additionally, the miR-3656 mimic decreased CUG2-induced cell migration, evasion, and sphere formation and sensitized the cells to doxorubicin. Suppression of JMJD5, with its small interfering RNA, impeded CUG2-induced cancer stem cell-like phenotypes. Thus, overexpression of CUG2 decreases miR-3656 levels, leading to upregulation of JMJD5, eventually contributing to cancer stem cell-like phenotypes.

## 1. Introduction

MicroRNAs (miRNAs) are endogenous noncoding RNAs of 19–22 nucleotides in length that negatively control the expression of target genes by inducing mRNA decay or translational suppression at the post-transcriptional level [1,2]. miRNAs are involved in various biological processes, including cell proliferation, differentiation, tumor formation, tumor suppression, and the resistance or sensitivity to anticancer drugs [1,3]. Thus, depending on the cellular context, miRNAs can act as oncogenes or tumor-suppressor genes. Matching of some bases of miRNAs (6–8 nucleotides) to the transcripts of tumor-suppressor genes and oncogenes can lead to the development of tumors and inhibition of tumorigenesis, respectively [4,5].

To identify an upregulated common transcript in various cancer tissues using the Affymetrix microarray system, increased transcripts of cancer upregulated gene (CUG) 2 have been detected in cancer tissues, including the lung, liver, ovary, and colon [6]. CUG2 exhibits oncogenic activities, such as by increasing the rate of cell proliferation and tumor formation in nude mice [6]. Moreover, cancer stem cell (CSC)-like phenotypes such as faster wound healing, aggressive cell invasion, enhanced sphere-forming ability, and increased doxorubicin-resistance have been attributed to CUG2 involving transforming growth factor-β signaling [7,8]. Both epidermal growth factor receptor/Stat1/HDAC4 and β-catenin/yes-associated protein/NIMA-related kinase 2 signal transduction are involved in these phenotypes [9,10]. Further studies demonstrated that c-Cbl decreased the levels of Spry2 protein when CUG2 was overexpressed, resulting in increased epidermal growth factor receptor and β-catenin protein levels and signaling [11]. A new component of kinetochore (CENP-W) forms a DNA-binding complex that interacts with the centromere-associated network component CENP-T, leading to proper cell mitosis and revealing that CENP-W corresponds to CUG2 [12]. CUG2 interacts with heterogeneous nuclear ribonucleoprotein U and enables attachment between kinetochore-microtubules in mitotic cells [13]. This effect facilitates chromatin transcription (FACT) complex, an H2A-H2B histone chaperone, and stabilizes the CENP-T/CUG2 complex at the centromere [14]. CUG2 also binds to and stabilizes EZH2, a catalytic subunit of polycomb recessive complex 2, leading to EZH2-mediated transcription repression [15].

JMJD5 carries a Jumonji C domain and possesses histone demethylase activity at the Lys36 position of histone 3, resulting in increased cyclin A expression that eventually contributes to increased cancer cell proliferation through epigenetic regulation [16,17]. JMJD5 associated with spindle microtubules participates in mitosis [18]. In addition, JMJD5 blocks p53 tumor suppressor activity by interacting with p53 protein, leading to reduced protein expression of p21, a cell cycle inhibitor [19]. JMJD5 also promotes the activity of hypoxia-inducible factor (HIF)-1α by binding to pyruvate kinase M2, promoting new blood vessel formation and cell growth under hypoxic conditions [20].

In this study, miRNA microarray analysis was performed using A549 and BEAS-2B cells stably expressing CUG2 to identify miRNAs involved in CUG2-induced CSC-like phenotypes. We found that miR-3656 levels were decreased in CUG2-overexpressing A549 and BEAS-2B cells compared to that in control cells, leading to elevated expression of JMJD5 protein, a candidate target of miR-3656. Administration of an miR-3656 mimic suppressed the CSC-like phenotypes under CUG2 overexpression by downregulating JMJD5; thus, miR-3656 mimics may be useful as therapeutic drugs against malignant cancers overexpressing JMJD5.

## 2. Materials and Methods

### 2.1. Cell Culture

MCF-7 and HeLa cells, purchased from the Korean Cell Line Bank (Seoul, Korea), were cultivated in Dulbecco’s Modified Eagle Medium. A549 and BEAS-2B cells, purchased from ATCC (Manassas, VA, USA), stably expressed CUG2 (A549-CUG2 or BEAS-CUG2) or contained an empty vector (A549-Vec or BEAS-Vec). A549-CUG2 and A549Vec cells were maintained in RPMI-1640, whereas BEAS-CUG2 and BEAS-Vec cells were maintained in Dulbecco’s modified Eagle medium supplemented with 10% fetal bovine serum, penicillin, and streptomycin under G410 at 500 μg/mL.

### 2.2. Antibodies and Transfection

Anti-JMJD5 antibodies were purchased from Abcam (ab106391; Cambridge, MA, USA), and actin antibodies were obtained from Santa Cruz Biotechnology (sc-47778; Santa Cruz, CA, USA). After reaching 70%–80% confluence, the cells were transfected with JMJD5 small interfering RNA (siRNA) or an miRNA-3656 mimic using Lipofectamine 2000 (Thermo Fisher Scientific, Waltham, MA, USA). Protein levels were measured at 40–48 h post-transfection by immunoblotting.

### 2.3. Immunoblotting

As previously described [7,8], proteins from the cell lysates were separated using a 10% sodium dodecyl sulfate–polyacrylamide gel, and the proteins on the gel were transferred onto nitrocellulose membranes. The membrane was treated with primary antibodies (1:500–1,000 dilution) followed by a horseradish peroxidase-conjugated secondary antibody. Images were obtained after treatment with electrochemiluminescence solution (Thermo Fisher Scientific, Carlsbad, CA, USA) using an ImageQuant LAS 4000 mini (GE Healthcare, Little Chalfont, UK).

### 2.4. Immunofluorescence Microscopy

As described previously [7], the expression of JMJD5 protein was examined in immunofluorescence analysis. The cells were fixed, permeabilized, and treated with anti-JMJD5 antibodies. An Alexa Fluor 488-conjugated secondary antibody was added to the treated cells and incubated for 1 h. Images were obtained under a fluorescence microscope after staining the cell nuclei with 4,6-diamidino-2-phenylindole.

### 2.5. 3′ Untranslated Region (UTR) Reporter Assay

Cells were transfected with wild-type (WT) or mutant JMJD5 3′-UTR-pmirGLO Dual-Luciferase reporter vectors together with the miR-3656 mimic or miR-control using Lipofectamine 2000 (Thermo Fisher Scientific, Carlsbad, CA, USA). Luciferase activity was measured at 2 days post-transfection and normalized to *Renilla* luciferase activity.

### 2.6. Wound Healing Assay

The cells were cultured until reaching approximately 80% confluence. Closure of the gap was measured by observing the cells under a light microscope for 24 h.

### 2.7. Transwell Invasion Assay

A transwell invasion assay was performed as described previously [7,8]. Briefly, cells in the upper chamber invaded the serum-containing lower chamber through a coated membrane (BD Biosciences, Franklin Lakes, NJ, USA). After fixing and staining the membrane with hematoxylin–eosin, images were obtained under a microscope (100× magnification). The experiment was conducted in triplicate. Data were expressed as the mean ± standard deviation (SD).

### 2.8. Sphere Forming Assay

For 6 days, cells treated with miRNAs or siRNAs were incubated in a medium containing 0.4% bovine serum albumin, 20 ng/mL epidermal growth factor, 10 ng/mL basic fibroblast growth factor, and 5 μg/mL insulin. Spheroids in wells with ultra-low attachment were examined to determine their size and number under an optical microscope. The assay was conducted in triplicate. Data represented the mean ± SD.

### 2.9. Measurement of Reactive Oxygen Species

Intracellular reactive oxygen species (ROS) levels in the cells were analyzed using a fluorescence microscope after treatment with 20 μM 2,7-dichlorodihydrofluorescein diacetate (Molecular Probes, Eugene, OR, USA) for 30 min.

### 2.10. Statistical Analysis

All data were presented as the mean ± SD. The results were analyzed using Student’s unpaired *t*-test for comparison between two groups. The data were considered as significant at a *p*-value of <0.05.

## 3. Results

### 3.1. JMJD5 Is a Functional Target of miR-3656 under CUG2 Overexpression

To investigate which miRNAs are involved in the development of CUG2-induced cancer, an miRNA array was conducted using A549 and BEAS-2B cells stably expressing CUG2 (A549-CUG2 and BEAS-CUG2, respectively), and the miRNAs were compared with those from control cells. Considering the abundance of miRNA and the same pattern (increase or decrease in miRNA levels in both A549-CUG2 and BEAS-CUG2 cells), we selected miR-3656, which was decreased in both A549-CUG2 and BEAS-CUG2 cells compared to that in their control cells (Appendix A). We found that the levels of miR-25-3p, miR-30a-5p, miR-100-5p, and miR-109-5p were increased in both A549-CUG2 and BEAS-CUG2 cells compared to that in the control cells; however, these results have been reported previously (Appendix A). Thus, we explored the role of decreased miR-3656 under CUG2 overexpression, as this result has not been reported in previous studies. TargetScan, a public bioinformatics analysis program, was used to identify target transcripts of miR-3656, revealing JMJD5 as a candidate target gene. Consistent with our hypothesis, A549-CUG2 and BEAS-CUG2 cells showed much higher levels of JMJD5 protein in the immunoblotting assay (Figure 1A) and immunofluorescence (Figure 1B) than the control cells, indicating that CUG2 overexpression upregulates JMJD5 protein expression when miR-3656 levels are decreased. A 3′UTR reporter assay was performed to confirm that the JMJD5 transcript is a target of the miR-3656 mimic. Using site-directed mutagenesis, we generated a 3′UTR mutant of the JMJD5 transcript to which the miR-3656 mimic could not bind, as shown in Figure 1C. As shown in Figure 1D, introduction of the miR-3656 mimic significantly reduced luciferase activity from the WT 3′UTR of the JMJD5 vector, whereas introduction of the miR-control did not exhibit this effect. Furthermore, unlike after miR-control treatment, introduction of the miR-3656 mimic failed to decrease luciferase activity from the 3′UTR mutant of the JMJD5 vector (Figure 1D). Administration of the miR-3656 mimic decreased JMJD5 protein levels in A549-CUG2 and BEAS-CUG2 cells (Figure 2A). The immunofluorescence assay confirmed that the miR-3656 mimic decreased JMJD5 protein levels (Figure 2B). In accordance with these results, MCF-7 and HeLa cells treated with the miR-3656 mimic showed decreased JMJD5 protein levels compared to cells treated with the miR control (Figure 2C). Treatment with the miR-3656 mimic also reduced JMJD5 transcript levels (Appendix A). Taken together, these results suggest that miR-3656 targets JMJD5 transcripts.

### 3.2. Administration of miR-3656 Mimic Hinders CUG2-Induced CSC-Like Phenotypes

As we observed decreased levels of miR-3656 in A549-CUG2 and BEAS-CUG2 cells, we examined whether the miR-3656 level is a critical factor determining the CUG2-induced CSC-like phenotypes, such as faster cell migration and invasion, increased sphere formation, and anti-cancer drug resistance. Compared with miR-control treatment, administration of the miR-3656 mimic delayed wound healing (Figure 3A) and cell invasion (Figure 3B) induced by CUG2. Moreover, treatment with the miR-3656 mimic lead to a reduced sphere size and number of spheres formed by CUG2 compared with miR-control transfection (Figure 3C). Our previous studies showed that under CUG2 overexpression, resistance to doxorubicin-induced cytotoxicity is closely related to a decrease in ROS production through upregulation of MnSOD [21]. Notably, introduction of the miR-3656 mimic enhanced ROS levels in A549-CUG2 and BEAS-CUG2 cells compared to miR-control treatment, indicating increased cytotoxicity following miR-3656 treatment (Figure 3D). Supporting this result, cleaved PARP, an indicator of apoptosis, was detected in the miR-3656 mimic-treated cells but not in the miR-control-treated cells (Appendix A). These results suggest that downregulation of miR-3656 levels is a critical factor in CUG2-induced CSC-like phenotypes.

### 3.3. Suppression of JMJD5, a Target of miR-3656, Inhibits CUG2-Induced CSC-Like Phenotypes

Given that the introduction of the miR-3656 mimic hampered CUG2-induced CSC-like phenotypes (Figure 3A–D), we examined whether suppression of JMJD5, a target of miR-3656, using its siRNA also inhibited CUG2-induced CSC-like phenotypes. We introduced JMJD5 siRNA into A549-CUG2 and BEAS-CUG2 cells to examine wound healing and cell invasion. As shown in Figure 4A,B, JMJD5 siRNA treatment inhibited CUG2-induced wound healing (Figure 4A) and cell invasion (Figure 4B) compared with control siRNA treatment. In addition, compared with control siRNA treatment, administration of JMJD5 siRNA reduced the number and size of spheres (Figure 4C). Furthermore, JMJD5 suppression drastically increased ROS production (enhanced cytotoxicity) in A549-CUG2 and BEAS-CUG2 cells treated with doxorubicin compared with control siRNA treatment (Figure 4D). Accordingly, PARP cleavage was detected in the cells treated with JMJD5 siRNA but not in the cells treated with control siRNA (Appendix A). On the basis of these results, upregulation of JMJD5 due to decreased miR-3656 levels is important for CUG2-induced CSC-like phenotypes.

## 4. Discussion

The analysis of potential target transcripts targeted by miR-3656 revealed CLCN6 (chloride channel, voltage-sensitive 6), paxillin, ADAMTS17 (ADAM metallopeptidase with thrombospondin type 1 motif, 17), c-Src, LTBP4 (latent transforming growth factor beta binding protein 4), JMJD5, and others. Among them, we excluded transcripts with unknown functions, those with well-known functions, or those unrelated to oncogenesis. Furthermore, proteins with low or inconsistent expression in A549-CUG2 and BEAS-CUG2 cells were excluded. For example, A549-CUG2 cells showed higher expression of paxillin compared to A549-Vec cells, whereas BEAS-Vec cells displayed upregulation of this protein compared to BEAS-CUG2 cells (Appendix A). LTBP4 protein was not expressed in A549-Vec and A549-CUG2 cells but was increased in BEAS-CUG2 cells compared to that in BEAS-Vec cells (Appendix A). Therefore, we focused on the role of JMJD5 in CUG2-induced oncogenesis.

Histone modification by methylation or demethylation has been reported as one of the mechanisms regulating protein expression [22]. JMJD5 uses H3K36me2 as a substrate, indicating that JMJD5 is a demethylase enzyme [17]. In addition, several lines of evidence have demonstrated that JMJD5 assists in the translocation of PKM2, thereby assisting in HIF-1α action under hypoxia. Furthermore, JMJD5 blocks the p53 tumor suppressor, suggesting that it functions as a proto-oncogene [19,20]. Elevated expression of JMJD5 has been detected in clinical breast cancer tissues and breast cancer cell lines such as MCF7 and MDA-MB231 cells, compared to that in paired adjacent normal mammary tissues and MCF-10A, a normal mammary epithelial cell line [23]. Studies reported that introducing the JMJD5-shRNA cassette into Caco2 cells using lentivirus or JMJD5 siRNA in oral squamous cell carcinoma reduced cell migration and invasion [24,25], supporting our finding that CSC-like phenotypes induced by CUG2 were suppressed when JMJD5 was inhibited. Moreover, JMJD5 is a target of the miR-3656 mimic; thus, treatment with the miR-3656 mimic is a potential therapeutic strategy for patients with cancers overexpressing JMJD5.

The functions of the miR-3656 mimic have not been well documented, despite a few recent studies. The levels of miR-3656 in circulating miRNAs were found to be lower in patients with early stage breast cancer than those in normal subjects [26]. In other studies, exosomes from human neural stem cells stimulated by interferon-γ were found to contain miR-3656, which improved cell survival in an ischemic stroke model [27]. These findings indicate that miR-3656 can be used as a biomarker of cancer or cell survival. A recent study showed that gemcitabine-resistant Panc-1 cells had lower miR-3656 levels than parental Panc-1 cells [28]. Subsequent research revealed that miR-3656 targets RHOF, a member of the Rho subfamily of GTPases, leading to inhibition of epithelial-mesenchymal transition [28]. In a hypertension model, miR-3656 suppressed cell proliferation and migration of human umbilical vein endothelial cells but induced apoptosis of these cells by targeting eNOS and ADAMTS13 [29]. Herein, we demonstrated that the miR-3656 mimic targets JMJD5 and that decreased levels of miR-3656 cause JMJD5, an upregulated proto-oncogene. In addition, treatment with the miR-3656 mimic decreased the expression of PKM2 protein, a binding partner of JMJD5 (Appendix A), resulting in an assignment to resolve.

Tumor tissue microarrays have been widely used for immunohistochemistry of paraffin sections, whereas miRNA arrays have not been used for this purpose. Thus, the same clinical samples should be newly prepared for both immunohistochemistry (paraffin section) and miRNA arrays to determine the relationship between CUG2-induced upregulation of JMJD5 and downregulation of miR-3656. Further studies are needed to validate our findings using the same tumor tissues prepared for detecting both JMJD5 protein and miR-3656 levels under overexpression of CUG2.

Recently, the molecular mechanism underlying the regulation of miRNA levels has been revealed, leading to the development of precise prediction models for the positioning of miRNA promoters [30,31]. Some transcription factors such as P53, RREB1, HIF1, and Myc increase or decrease the expression of specific miRNAs [30,32,33,34,35]. Epigenetic alterations in miRNA promoters, such as hypermethylation and CpG methylation, also affect miRNA expression levels [30,36,37,38]. Thus, studies are needed to identify the specific transcription factors or epigenetic alterations in the miRNA promoter involved in decreasing miR-3656 levels under CUG2 overexpression.

## 5. Conclusions

Overexpression of CUG2 decreases miR-3656 levels, leading to upregulation of JMJD5, eventually contributing to cancer stem cell-like phenotypes. In addition, treatment with the miR-3656 mimic inhibits the potential oncogenic roles of JMJD5 in cell migration and invasion and anticancer drug resistance, providing a potential therapeutic biological drug against malignant cancers overexpressing JMJD5.

## Figures and Tables

**Figure 1 biomolecules-12-00122-f001:**
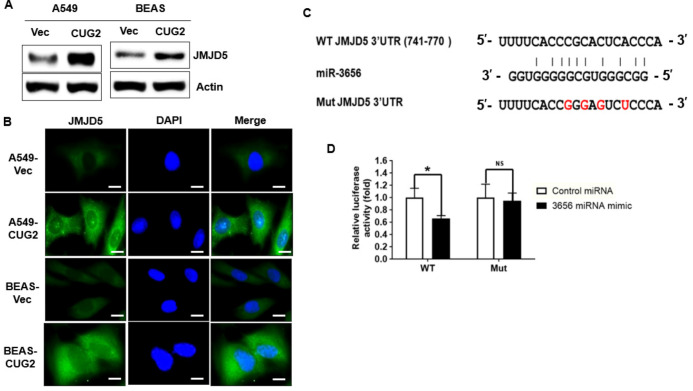
CUG2 downregulates miR-3656, which targets the JMJD5 transcript. (**A**,**B**) JMJD5 protein levels in A549-CUG2, BEAS-CUG2, and control cells were measured using immunoblotting assay and under immunofluorescence microscopy. Assays were repeated twice. Scale bar indicates 10 μm. 4,6-diamidino-2-phenylindole (DAPI) was used to stain the nucleus; (**C**) Predicted miR-3656 target sequences in JMJD5 3′UTR and mutated nucleotide sequence of JMJD5 3′UTR (**D**) HEK293A cells were transfected with the miR-3656 mimic and luciferase reporter vectors containing either the WT or mutant nucleotide sequence of JMJD5 3′UTR. The assay was conducted in triplicate, and error bars indicate the SD. (* *p* < 0.05; miR-3656 vs. miR-control in WT 3′UTR of JMJD5). NS, not significant.

**Figure 2 biomolecules-12-00122-f002:**
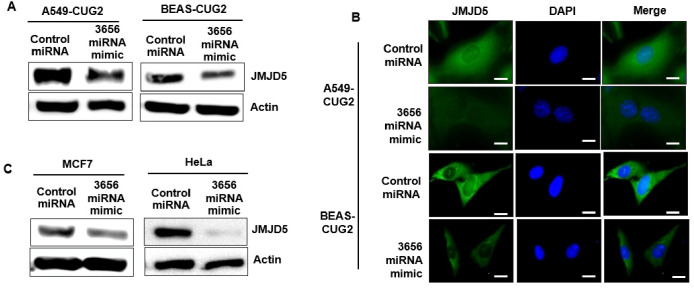
Administration of miR-3656 mimic decreased JMJD5 expression protein under CUG2 overexpression. (**A**,**B**) JMJD5 protein levels in A549-CUG2 and BEAS-CUG2 cells were measured using immunoblotting and immunofluorescence at 2 days post-transfection with the miR-3656 mimic or control miRNA. Scale bar indicates 10 μm. Assays were repeated twice; (**C**) JMJD5 protein levels in HeLa and MCF7 cells were measured by immunoblotting at 2 days post-transfection with the miR-3656 mimic or control miRNA.

**Figure 3 biomolecules-12-00122-f003:**
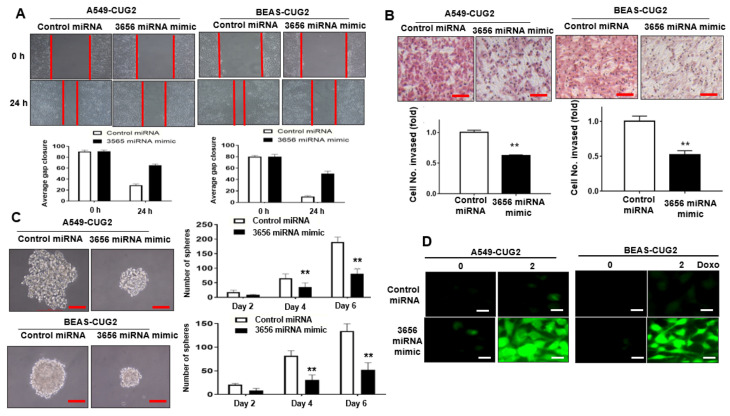
Treatment with miR-3656 mimic diminished CUG2-induced CSC-like phenotypes. (**A**) Migration of A549-CUG2 and BEAS-CUG2 cells treated with the miR-3656 mimic or miR-control was measured in a wound healing assay; (**B**) Cell invasion by A549-CUG2 and BEAS-CUG2 cells treated with the miR-3656 mimic or miR-control was measured in a transwell invasion assay. The assay was performed in triplicate, and the error bars indicate the SD. Scale bar indicates 10 μm. (** *p* < 0.01, miR-3656 mimic vs. miR-control); (**C**) A549-CUG2 and BEAS-CUG2 cells were treated with the miR-3656 mimic or miR-control. Spheroid size and number were evaluated for 6 days post-seeding. Spheroid size >50 mm was the criterion for sphere formation. The assay was conducted in triplicate, and the error bars indicate the SD. Scale bar indicates 10 μm. (** *p* < 0.01, miR-3656 mimic vs. miR-control); (**D**) Doxorubicin was treated for 12 h after transfection of A549-CUG2 and BEAS-CUG2 cells with the miR-3656 mimic or miR-control. ROS were detected in the cells under a fluorescence microscope using 2,7-dichlorodihydrofluorescein diacetate (20 μM). Scale bar indicates 10 μm.

**Figure 4 biomolecules-12-00122-f004:**
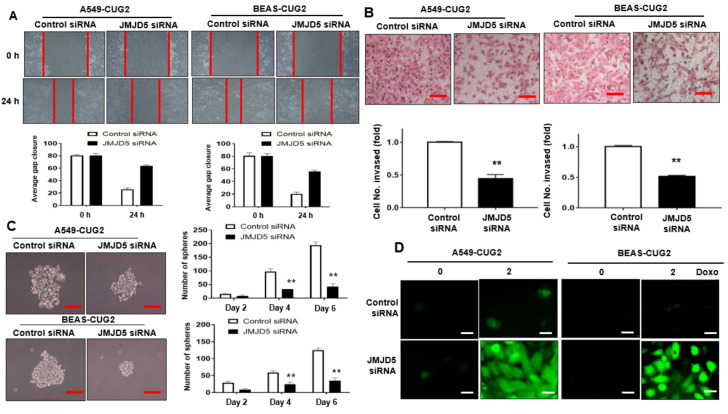
Administration of JMJD5 siRNA hampered CSC-like phenotypes under CUG2 overexpression. (**A**) Cell migration from A549-CUG2 and BEAS-CUG2 cells treated with JMJD5 siRNA or control siRNA was measured in a wound healing assay; (**B**) Cell invasion from A549-CUG2 and BEAS-CUG2 cells treated with JMJD5 siRNA or control siRNA was measured in a transwell invasion assay. The assay was performed in triplicate, and error bars indicate the SD (***p* < 0.01, JMJD5 siRNA vs. control siRNA); (**C**) A549-CUG2 and BEAS-CUG2 cells were treated with JMJD5 siRNA or control siRNA. Spheroid size and number were evaluated for 6 days post-seeding. A spheroid size >50 mm was the criterion for sphere formation. The assay was conducted in triplicate, and error bars indicate the SD (** *p* < 0.01, miR-3656 mimic vs. miR-control); (**D**) Doxorubicin was treated for 12 h after transfection of A549-CUG2 and BEAS-CUG2 cells with JMJD5 siRNA or control siRNA. ROS were detected in the cells under a fluorescence microscope using 2,7-dichlorodihydrofluorescein diacetate (20 μM). Scale bar indicates 10 μm.

## Data Availability

Data can be provided from the corresponding author upon reasonable request.

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
