# Peer review of "Elevated Expression of JMJD5 Protein Due to Decreased miR-3656 Levels Contributes to Cancer Stem Cell-Like Phenotypes under Overexpression of Cancer Upregulated Gene 2"

_biomolecules, 2022, doi:10.3390/biom12010122_

Round 1

Reviewer 1 Report

The manuscript by Yawut and colleagues describes a possible mechanism of tumor progression in CUG2 overexpressing cancers.

Indeed, to elucidate the molecular bases of CUG2-induced oncogenesis, miRNAs expression was evaluated in cell lines highlighting a significantly down-expressed miRNA. Next, a target of this miRNA was found to be highly expressed and to exert a role in cells transformation. Even if further studies are required to validate these results especially in tumor tissues from patients, this result may open the way for advancements in this field. These limitations should be clearly reported in the discussion section.

Moreover, within the results, some points have to be better clarified. Is the miR-3656 the only one found significantly down expressed between the A549-CUG2 and BEA-CUG2 and the respective control cells? If yes, as Supplementary table 1 suggests, it has to be better clarified in the text. Next, 4 miRNAs are reported to be significantly up-expressed. What about these miR? Why they were not further explored? Finally, among the miR-3656 predicted targets, why the JMJD5 was chosen? The other targets and the reasons why they were not explored have to be reported.

Minor points:

line 43: delete "they";

line 139: 3-fold higher..."

Author Response

We are grateful for the careful and keen comments from the reviewers on our manuscript (Ref biomolecules-1531123; Elevated expression of JMJD5 protein due to decreased miR-3656 levels contributes to cancer stem cell-like phenotypes under overexpression of cancer upregulated gene (CUG)2). We have carefully considered the expert comments and suggestion to improve the quality of our manuscript. We hope that the revised version of manuscript meets the standard required for publication in Biomolecules.

Q1: Reviewer#1 asked to discuss necessity of clinical results to validate our findings: upregulation of JMJD5 due to decreased miR-3656 under CUG2 overexpression.

As suggested, we discussed necessity of clinical results to validate our findings in the Discussion part.

In many cases, tumor tissue microarray has been prepared for immunohistochemistry using paraffin sections but not for miRNA array. Thus, the same clinical samples should be newly prepared for the purpose of both immunohistochemistry (paraffin section) and miRNA array to demonstrate the relationship between CUG2-induced upregulation of JMJD5 and downregulation of miR-3656. We remain further studies that should validate our findings using the same tumor tissues to be prepared for detection of both JMJD5 protein and miR-3656 levels under overexpression of CUG2.

Q2: Reviewer#1 asked why the authors pursuit a role of miR-3656 in CUG2 overexpression and why the authors focus on JMJD5 against miR-3656 as a target transcript.

We described the reason selecting miR-3656 in the Result section and JMJD5 as a target transcript against miR-3656 in the Discussion section.

- The reason selecting miR-3656 in the Result section

Considering abundance of miRNA, and the same pattern (increase or decrease of miRNA levels in both A549-CUG2 and BEAS-CUG2 cells), we selected miR-3656 that was decreased in both A549-CUG2 and BEAS-CUG2 cells compared to their control cells. (Supplementary Figure 1). In addition, although we selected miR-25-3p, miR-30a-5p, miR-100-5p, and miR-109-5p levels that were increased in both A549-CUG2 and BEAS-CUG2 cells compared with those in the control cells, they were lack of novelty (Supplementary Figure 1). We thus explored a role of the decreased miR-3656 under CUG2 overexpression.

- The reason selecting JMJD5 as a target transcript against miR-3656 in the Discussion section

Regarding potential target transcripts against miR-3656, TargetScan program showed CLCN6 (chloride channel, voltage-sensitive 6), paxicillin, ADAMTS17(ADAM metallopeptidase with thrombospondin type 1 motif, 17), c-Src, LTBP4 (latent transforming growth factor beta binding protein 4), JMJD5, and so on. Among them, we first excluded transcripts with unknown function, with well-known function or unrelated to oncogenesis. Furthermore, proteins with low or inconsistent expression in A549-CUG2 and BEAS-CUG2 cells were excluded. For example, A549-CUG2 cells showed higher expression of paxillin than A549-Vec cells while BEAS-Vec cells displayed upregulation of this protein compared to BEAS-CUG2 cells (Supplementary Figure 4). Western blotting could not display LTBP4 protein levels in both A549-Vec and A549-CUG2 cells whereas LTBP4 expression was increased in BEAS-CUG2 cells compared to that in BEAS- Vec cells (Supplementary Figure 4). Therefore, we focused on a role of JMJD5 in CUG2-induced oncogenesis.

Q3. The reviewer #1 point out typo-errors.

We fixed the typo-errors as mentioned.

Reviewer 2 Report

In this manuscript Yawut et al. propose a role for miR-3656 and JMJD5 in cancer stem cell-like phenotypes in the context of CUG2 upregulation. Authors quite convincingly demonstrate that miRNA-3656 targets JMJD5 expression, and that miR-3656 downregulation upon CUG2 overexpression promotes stem-cell-like phenotypes via JMJD5 upregulation. However, before publication can be recommended, it would be important to further discuss the link between the identified axis and stem cell-like phenotypes, as well as the mechanisms that lead to miR-3656 downregulation upon elevated levels of CUG2. Also, some observations remain superficially discussed. Specific points that deserve additional attention are outlined below:

  • In this work authors focus on the mechanisms that lead to cancer stem cell-like phenotypes upon CUG2 (cancer upregulated gene 2) overexpression. The official gene symbol approved by the HGNC for CUG2 is CENPW (Centromere protein W), and therefore, authors should, at least, introduce this name in the manuscript. Moreover, Introduction section needs to be improved, including a rationale for assessing CUG2-mediated miRNA expression and a more thorough description of CENPW in transcriptional regulation, including important references that show a link between CENPW and transcriptional regulators such as the FACT complex (Genes Dev. 2016;30(11):1313–1326) and EZH2 (Biochem Biophys Res Commun. 2015;464(1):256–262.)
  • Microarray results are shown insufficiently in the manuscript, as only a supplementary table with microRNAs differentially expressed by 3-fold are shown. Relative miRNA expression profiles of all differentially expressed miRNAs should be provided, including the statistical significance, for example in a Heatmap representation. Moreover, how the microarray was performed, how many samples were used, and the statistical analysis applied for the differential expression analysis should be described in the Materials & Methods section.
  • Authors focus on JMJD5 as a putative target of miR-3656, however, I would suggest to include all the putative targets found by TargetScan analysis, and explain why authors decided to focus on JMJD5. In this regard, it would be important to further discuss how JMJD5 might be involved in stem cell-like phenotype development, including references of works linking JMJD5-mediated gene regulation with cancer-promoting phenotypes.
  • In the immunofluorescence analyses shown in Figure 1 and 2, JMJD5 localization seems to be both nuclear and cytoplasmic. This result seems odd, considering that JMJD5 localization has been shown to be mainly nuclear. How convinced are authors about the specificity of these immunofluorescence experiments?
  • Authors use mainly lung cancer cell lines for their studies, it would be important to explain the reason behind the cell model of choice. Also, in the Materials and Methods section the HEK293 cells used for luciferase assays should be included.
  • In Supplementary Figure 1 Q-PCR data is represented as “Relative intensity”, this is not a usual way to represent gene expression data. I would suggest to use the term “Relative mRNA levels” instead.
  • In the Materials and Methods section all the antibodies used in the work should be included. Cleaved PARP1 and PKM2 antibodies are missing. Regarding PARP1 analysis, does the used antibody  recognize exclusively the cleaved version of PARP1?
  • Authors should perform a thorough proofread on the manuscript in order to avoid typos.

Author Response

We are grateful for the careful and keen comments from the reviewers on our manuscript (Ref biomolecules-1531123; Elevated expression of JMJD5 protein due to decreased miR-3656 levels contributes to cancer stem cell-like phenotypes under overexpression of cancer upregulated gene (CUG)2). We have carefully considered the expert comments and suggestion to improve the quality of our manuscript. We hope that the revised version of manuscript meets the standard required for publication in Biomolecules.

Q1. Reviewer#3 suggested to introduce other functions of CENP-W (another name of CUG2) with cell cycle aspects in the Introduction section.

As suggested, we introduced other functions of CENP-W in the Introduction section.

Meanwhile, a new component of kinetochore (CENP-W) forms a DNA-binding complex interacted with centromere-associated network (CCAN) component CENP-T leading to performing a proper cell mitosis, which reveals that CENP-W corresponds to CUG2 [12]. CUG2 interacts with heterogeneous nuclear ribonucleoprotein U and confers attachment between kinetochore-microtubule in mitotic cells [13]. The facilitates chromatin transcription (FACT) complex, a H2A-H2B histone chaperone, stabilizes CENP-T/CUG2 complex at centromere [14]. In addition, as another function of CUG2, CUG2 binds to EZH2, a catalytic subunit of polycomb recessive complex 2(PRC2), which subsequently stabilizes EZH2 protein, leading to EZH2-mediated transcription repression [15].

Q2. Reviewer #3 asked heat map of miRNAs regarding A549-Vec/-CUG2 and BEAS-Vec/-CUG2 cell lines.

As suggested, we analyzed miRNAs from regarding A549-Vec/-CUG2 and BEAS-Vec/-CUG2 cell lines with help from Macrogen. We added this result in Supplementary Figure 1.

Q3. Reviewer #3 also asked other putative targets for miR-3656 in addition of JMJD5 and reason for selection of JMJD5.

As suggested, we described this issue in the Discussion section.

Regarding potential target transcripts against miR-3656, TargetScan program showed CLCN6 (chloride channel, voltage-sensitive 6), paxicillin, ADAMTS17(ADAM metallopeptidase with thrombospondin type 1 motif, 17), c-Src, LTBP4 (latent transforming growth factor beta binding protein 4), JMJD5, and so on. Among them, we first excluded transcripts with unknown function, with well-known function or unrelated to oncogenesis. Furthermore, proteins with low or inconsistent expression in A549-CUG2 and BEAS-CUG2 cells were excluded. For example, A549-CUG2 cells showed higher expression of paxillin than A549-Vec cells while BEAS-Vec cells displayed upregulation of this protein compared to BEAS-CUG2 cells (Supplementary Figure 3). Western blotting could not display LTBP4 protein levels in both A549-Vec and A549-CUG2 cells whereas LTBP4 expression was increased in BEAS-CUG2 cells compared to that in BEAS- Vec cells (Supplementary Figure 3). Therefore, we focused on a role of JMJD5 in CUG2-induced oncogenesis.

Q4. Reviewer #3 asked further discussion how JMJD5 might be involved in stem cell –like phenotypes.

Using key words such as JMJD5 and cancer, pubmed displayed only 25 papers until now . There are not many literatures regrading JMJD5 and cancer.

As we already mentioned several lines of evidence in this manuscript for emphasizing JMJD5’s role in cancer stem cell-like phenotypes, we added another reference showing that JMJD5 regulates p53/NF-κB pathway for apoptosis and metastasis.

Reviewer 3 Report

In this paper, Yawut et al. describes a novel regulatory target of miR-3656 that may impact CUG2-induced cancer stem cell-like phenotype. The authors used various overexpression and silencing approaches to claim that miR-3656 regulate JMJD5 protein, they further conclude by either using miR-3656 mimic or JMJD5 siRNA decreased CUG2-induced cell migration, evasion, and sphere formation in A549 and BEAS-2B cell line. Overall, an interesting paper, the experiments were carefully designed and executed. The techniques used are appropriate to the questions being addressed. I have few minor comments for the authors to take into consideration before publication.

  1. The authors should at least mention in legend section for each figure the number of times individually the experiments were repeated especially for western blot and microscopy result since none of them have quantification.
  2. Also, it would be better to have scale bar for microscopy images.
  3. The authors should check the manuscript for typos.

Author Response

We are grateful for the careful and keen comments from the reviewers on our manuscript (Ref biomolecules-1531123; Elevated expression of JMJD5 protein due to decreased miR-3656 levels contributes to cancer stem cell-like phenotypes under overexpression of cancer upregulated gene (CUG)2). We have carefully considered the expert comments and suggestion to improve the quality of our manuscript. We hope that the revised version of manuscript meets the standard required for publication in Biomolecules.

Q1. Reviwer#2 asked to describe performing times for qualitative experiments such as Western blotting and Microscopy result in figure legend

As suggested, we mentioned the number of times for experiments in legend section.

Q2. Reviwer#2 asked to add scale bar in microscopy images.

As pointed out, we added scale bar in microscopy images.

Q3. The reviewer #1 point out typo-errors.

We fixed the typo-errors as mentioned.